# Macropinocytosis in Different Cell Types: Similarities and Differences

**DOI:** 10.3390/membranes10080177

**Published:** 2020-08-03

**Authors:** Xiao Peng Lin, Justine D. Mintern, Paul A. Gleeson

**Affiliations:** The Department of Biochemistry and Molecular Biology and Bio21 Molecular Science and Biotechnology Institute, The University of Melbourne, Melbourne 3010, Victoria, Australia; xiaol1@student.unimelb.edu.au (X.P.L.); jmintern@unimelb.edu.au (J.D.M.)

**Keywords:** macropinocytosis, macropinosomes, membrane ruffling, dendritic cells, macrophages, endothelial cells, epithelial cells, neurons, microglia, cancer cells, amiloride

## Abstract

Macropinocytosis is a unique pathway of endocytosis characterised by the nonspecific internalisation of large amounts of extracellular fluid, solutes and membrane in large endocytic vesicles known as macropinosomes. Macropinocytosis is important in a range of physiological processes, including antigen presentation, nutrient sensing, recycling of plasma proteins, migration and signalling. It has become apparent in recent years from the study of specialised cells that there are multiple pathways of macropinocytosis utilised by different cell types, and some of these pathways are triggered by different stimuli. Understanding the physiological function of macropinocytosis requires knowledge of the regulation and fate of the macropinocytosis pathways in a range of cell types. Here, we compare the mechanisms of macropinocytosis in different primary and immortalised cells, identify the gaps in knowledge in the field and discuss the potential approaches to analyse the function of macropinocytosis in vivo.

## 1. Introduction

Endocytosis is a process carried out by eukaryotic cells that facilitates the internalisation of extracellular molecules and plasma membrane components. Over the past 15–20 years, it has become clear that there are many distinct pathways of endocytosis that promote the internalisation of a range of extracellular molecules, either via specific or nonspecific mechanisms [1]. Macropinocytosis is one of the nonspecific endocytic pathways that has gained considerable recent attention, as this pathway has the unique property of rapidly internalising very large amounts of plasma membrane and extracellular fluid and functions in a range of physiological processes including nutrient uptake and nutrient sensing, signalling, antigen presentation and cell migration [2,3,4,5]. Macropinocytosis can also be exploited by pathogens such as bacteria, viruses, protozoa and prions to invade host cells [2,6]. Although macropinocytosis has only recently attracted attention in mainstream biology, Warren Lewis first described the morphological characteristics of this pathway in the 1930s, when he showed that macrophages and cancer cells ruffle and consequently internalise extracellular fluid [7,8].

Unlike other endocytic pathways such as receptor-mediated endocytosis and phagocytosis, macropinocytosis is not initiated by binding of cargo molecules to cognate cell surface receptors or contact of large particles with the cell surface, and is therefore not saturable [5]. Instead, macropinocytosis is initiated by the polymerisation of actin at the plasma membrane to generate extensions called membrane ruffles [2]. These ruffles may originate from the leading edge of cells (peripheral membrane ruffles) and fold backwards or from the dorsal surface of cells (dorsal membrane ruffles) to give rise to circular cups [9]. Ruffles can trap solutes into large (0.2–5 μm in diameter), irregular-shaped vesicles called macropinosomes [2]. Macropinosomes can subsequently undergo homotypic fusion and fission and also traffic to other organelles in the endolysosomal system [10]. 

Much of the early work on macropinocytosis was focused on the immune cells, dendritic cells and macrophages. However, the range of cell types capable of macropinocytosis is now recognised to include not only dendritic cells and macrophages, but also B and T cells, epithelial and endothelial cells, fibroblasts, neurons, microglia and cancer cells. While some of these cell types only show appreciable macropinocytosis in response to growth factor stimulation or exposure to pathogens, for example epithelial cells [11], other cell types can macropinocytose constitutively, such as immature dendritic cells [12,13,14], macrophages [15] and cancer cells [16]. As macropinocytosis is the most efficient way for cells to internalise large amounts of extracellular fluid and plasma membrane, this pathway has a significant impact on the physiology of these cells.

It is becoming increasingly apparent that macropinocytosis is not identical in different cell types; rather, macropinocytosis is regulated by different stimuli and unique molecular machinery in specific cell types. For example, sorting nexin 5 (SNX5) has been identified as a critical regulator of dorsal ruffling and macropinocytosis in macrophages [9,17] but does not affect macropinocytosis in splenic dendritic cells [9]. Moreover, the fate of macropinosome maturation also varies between different cells; in some cases, macropinosomes are transported along the endosomal–lysosomal pathway [18]; and in other cases, macropinosomes are recycled back to the PM [15]. Furthermore, the same molecular machinery can be utilised for different aspects of macropinocytosis in different cell types. For example, phosphoinositide 3-kinase (PI3K) regulates macropinocytosis by promoting macropinosome closure in macrophages [19] but regulates membrane ruffling in endothelial cells [20]. In addition, the Na^+^/H^+^ exchanger inhibitor amiloride robustly inhibits macropinocytosis in macrophages [21], whereas it does not inhibit macropinocytosis in dendritic cells [22].

The finding that there are differences in the macropinocytosis pathways in different cell types highlights the importance of interrogating macropinocytosis in a range of specialised cell types, particularly in physiologically relevant primary cells. There have been a number of reviews on macropinocytosis which have provided an excellent integrated critique of this pathway [2,3,4,5]. This review aims to focus on the current status of our understanding of macropinocytosis in primary cells, to compare the molecular mechanisms of this pathway between the different cell types, and to highlight the gaps in knowledge.

### Analysing Macropinocytosis

Despite its importance to physiology, the molecular mechanisms underlying macropinocytosis remain only partly understood. This is largely due to the difficulty in studying macropinocytosis. Unlike clathrin-coated vesicles derived from clathrin-mediated endocytosis, macropinosomes have no apparent coat structure [2] and currently no unique molecules present on macropinosome membranes have been identified. Instead, macropinosomes are typically labelled with fluorescently-tagged fluid-phase markers that are known to be predominantly internalised by macropinocytosis, such as dextran, Lucifer yellow, albumin, ovalbumin and horseradish peroxidase (HRP) [10,12,23,24]. Labelling with fluid-phase markers has the limitation in that uptake by other endocytic processes can also contribute. For example, dextran, ovalbumin and HRP are also internalised by mannose receptor-mediated endocytosis [12,25]. Dextran is one of the most commonly used macropinosome markers, and a range of dextran sizes have been used to identify macropinosomes, typically between 3 and 70 kDa. Further, 70 kDa dextran has been shown to be more selective for labelling of macropinosomes compared with 10 kDa dextran [26] and, perhaps not surprisingly, (micro)pinocytosis has been shown to make a significant contribution to the uptake of smaller 3 kDa dextran [19]. These limitations in the specificity of fluid-phase markers for macropinocytosis highlight the need to identify macropinosome-specific molecules.

An additional difficulty in studying macropinocytosis is the lack of drugs that specifically inhibit this process [27]. Drugs that have been commonly used for this purpose target actin polymerisation, which is required for macropinocytosis. These include cytochalasins, which bind to actin filaments and prevent their polymerisation [28,29] and latrunculins, which bind to and sequester monomeric actin [29,30]. PI3K inhibitors such as wortmannin and LY294002 are often used to inhibit macropinocytosis through the inhibition of actin polymerisation [28]. However, these drugs are not specific for macropinocytosis, as other endocytic pathways such as phagocytosis also require actin polymerisation [27]. To date, the most selective inhibitors of macropinocytosis are members of the amiloride family, including amiloride and its analogues 5-(*N*-ethyl-*N*-isopropyl)amiloride (EIP-amiloride) and 5-(*N,N*-dimethyl)amiloride (dimethyl amiloride), which inhibit the activity of Na^+^/H^+^ exchangers [11,30]. Inhibition of Na^+^/H^+^ exchangers leads to acidification of the submembranous cytosol with a subsequent failure in recruitment of the small GTPases Rac1 and Cdc42 to the plasma membrane, and the inhibition of actin polymerisation [30]. However, as amiloride and its analogues have effects on other cellular processes, they cannot be considered specific inhibitors of macropinocytosis [31].

## 2. Non-Mammalian Systems

Macropinocytosis has been studied in a number of non-mammalian organisms. These include the amoebae *Dictyostelium*, where macropinocytosis facilitates the acquisition of nutrients for their growth in liquid culture [32]. Studies in *Dictysostelium* have revealed the requirement for the phosphoinositide PI(3,4,5)P_3_ and the small GTPases Ras and Rac, which are recruited to patches of the plasma membrane [33]. Furthermore, using lattice light-sheet microscopy (LLSM), Veltman and colleagues demonstrated that the actin-driven cup projections in *Dictyostelium* formed around these patches of PI(3,4,5)P_3_, Ras and Rac [34]. *Caenorhabditis elegans* has also been a useful model to study macropinocytosis in vivo, as scavenger cells known as coelomocytes can non-specifically internalise fluid-phase markers such as 40 kDa dextran and albumin injected into its body cavity [35]. The ability to visualise the internalisation of injected dextran into the body cavity, as well as the ability to readily genetically manipulate these organisms, has allowed the interrogation of various aspects of macropinocytosis such as the trafficking of different-sized dextrans [26] and genes involved in phosphoinositide metabolism [36]. When injected into the embryo, *Drosophila melanogaster* hemocytes are capable of internalising 70 kDa dextran into large vesicles (typically 1–4 μm in diameter), suggesting that these cells are also capable of macropinocytosis [37]. Given these features, *Drosophila* has considerable potential as a model system to define the mechanisms of macropinocytosis in whole organisms.

## 3. Mammalian Cells

### 3.1. Immune Cells

#### 3.1.1. Dendritic Cells

Dendritic cells (DCs) initiate immune responses through the presentation of peptide–MHC complexes at their surface, leading to recognition of the peptide antigen–MHC complex and activation of T cells [38]. The ability of DCs to efficiently present peptide–MHC complexes is in part attributed to their ability to efficiently capture antigens by macropinocytosis [12,14]. DCs, in addition to macrophages, are the most well-studied cell types with respect to macropinocytosis. Distinct subclasses of DCs are identified under steady-state conditions, namely conventional DCs (cDCs) and plasmacytoid DCs (pDCs) [39]. cDCs can be further subdivided into cDC1 and cDC2 populations, which show differences in the pathways of antigen presentation resulting in different downstream T cell responses [39]. Several DC populations have been used to study macropinocytosis. These include primary DCs isolated from mouse spleen [39] and DCs derived from the differentiation of DC precursors from mouse bone marrow with GM-CSF (bone marrow-derived dendritic cells (BMDCs)) [40]. Human monocyte-derived DCs have also been used as a DC model [41].

##### Role of Macropinocytosis in Antigen Presentation by DCs

Macropinocytosis is utilised by DCs to facilitate uptake of extracellular antigens, a process called antigen capture, for subsequent intracellular processing of antigens into antigenic peptides, which is required for generation of peptide–MHC complexes. Type II collagen (CII) is a long (~300 nm) autoantigen molecule implicated in collagen-induced arthritis in mice and rheumatoid arthritis in humans and is internalised by macropinocytosis in mouse BMDCs [42]. Importantly, amiloride blocks presentation of CII antigen to T cells both in vitro and in vivo, highlighting the therapeutic potential of blocking macropinocytosis in the context of autoimmune diseases [42]. Uptake of RNA by macropinocytosis has also been demonstrated in human monocyte-derived DCs, a finding relevant to the development of antigen-encoding RNA vaccines [43]. Evidence for the role of macropinocytosis in DC antigen presentation is also revealed through the analysis of the trafficking of fluid-phase markers. In human monocyte-derived DCs, internalised 40 kDa dextran is initially localised to macropinosomes in the periphery of the cell and subsequently trafficked to MHC II-expressing lysosomes, demonstrating that macropinosome content is delivered to MHC II-loading compartments in DCs [12] (Figure 1A). The process of macropinocytosis, and associated plasma membrane turnover, is also proposed to facilitate uniform migration of DCs in their environment, thus allowing them to function as efficient antigen sampling cells [44].

##### Constitutive Macropinocytosis in DCs and the Effect of Activating Stimuli

In contrast to many other cell types, macropinocytosis in DCs is a constitutive process, which enables them to function as sentinel cells that constitutively capture antigens [12,13,14]. Constitutive macropinocytosis in the absence of defined stimulatory factors has been demonstrated for both mouse BMDCs [14] and mouse splenic DCs [13]. Constitutive macropinocytosis in human monocyte-derived DCs depends on extracellular calcium and the calcium-sensing receptor (CaSR), a G protein-coupled receptor [45]. Furthermore, plasma membrane-localised phosphatidic acid is required for constitutive membrane ruffling and macropinocytosis in mouse BMDCs [46]. The role of extracellular calcium, CaSR and phosphatidic acid in constitutive macropinocytosis is described in more detail below for macrophages.

The activation status of DCs is considered to regulate their ability to macropinocytose. In most tissues, DCs are present in an “immature” state, specialising in antigen capture by endocytic processes such as macropinocytosis [12,14]. Following antigen capture, DCs become activated or “mature”, migrating to T cell-dependent areas of secondary lymphoid organs and losing their ability to efficiently acquire antigens [12,38]. For example, immature BMDCs can internalise ovalbumin whereas mature BMDCs show no uptake [47]. It has been suggested that antigen-presenting mature DCs may downregulate macropinocytosis as any additional uptake of antigens is not necessary and, furthermore, macropinocytosis may interfere with antigen presentation as a consequence of promoting internalisation of peptide–MHC complexes from the cell surface [48].

##### Molecular Machinery Involved in Macropinocytosis by DCs

The small GTPases Rac1 and Cdc42, which are involved in actin polymerisation [49], are key regulators of constitutive macropinocytosis in DCs. Microinjection of dominant-negative Rac1 or dominant-negative Cdc42 into immature BMDCs inhibits macropinocytosis and microinjection of constitutively active Rac1, or constitutively active Cdc42, increases macropinocytosis in mature BMDCs [47]. Similar findings have been reported for macropinocytosis in splenic DCs; microinjection of dominant-negative forms of Rac1 or Cdc42 leads to a decrease in macropinocytosis [13]. As neither dominant-negative Rac1 nor dominant-negative Cdc42 inhibit membrane ruffling in splenic DCs, it is likely that Rac1 and Cdc42 regulate macropinocytosis downstream of membrane ruffling [13].

The effect of inhibiting other cellular machinery on macropinocytosis in DCs has also been evaluated. Inhibition of PI3K with wortmannin in splenic DCs inhibits macropinocytosis without inhibiting membrane ruffling, suggesting that similar to Rac1 and Cdc42, the regulation of macropinocytosis by PI3K in DCs occurs downstream of membrane ruffling [13]. Rapamycin inhibits macropinocytosis in BMDCs and splenic DCs, suggesting that mTOR signalling regulates macropinocytosis in DCs [50]. Inhibition of aquaporins also decreases macropinocytosis in human monocyte-derived DCs [51]. The glucocorticoid-induced leucine zipper protein (GILZ) is also a regulator of macropinocytosis in DCs [52]. GILZ negatively regulates macropinocytosis, although this varies depending on the DC subset; GILZ inhibits macropinocytosis in splenic cDC1 and BMDCs but not splenic cDC2 and splenic pDCs [52].

#### 3.1.2. Macrophages

Macrophages are key innate immune cells that have various functions including phagocytosis, antigen presentation, tissue repair and cytokine production [53]. Macropinocytosis is important for a number of these functions. Our understanding of the different pathways of macropinocytosis, such as constitutive versus growth factor-induced macropinocytosis, and dorsal versus peripheral ruffling pathways of macropinocytosis, largely result from the studies using macrophages. Macrophages from a variety sources have been used to study macropinocytosis, in particular primary mouse bone marrow-derived macrophages (BMDMs) [9,17], primary mouse peritoneal macrophages [17], the macrophage cell line RAW 264.7 [54,55] and human monocyte-derived macrophages (hMDMs) [45,56].

##### Constitutive and Activated Macropinocytosis in Macrophages

Macropinocytosis occurs constitutively in macrophages [15] and is enhanced in response to growth factors such as macrophage colony-stimulating factor (CSF-1) [45] and to pathogen-derived molecules such as lipopolysaccharide (LPS) [54]. Constitutive macropinocytosis generates smaller macropinosomes than those arising after cell stimulation [45], whereas CSF-1 [45] and LPS [54] stimulation results in the formation of larger macropinosomes.

The molecular basis of constitutive macropinocytosis in macrophages has recently been elucidated [45]. Constitutive macropinocytosis, but not growth factor- or pathogen-induced macropinocytosis, is dependent on extracellular calcium and CaSR, which is expressed in myeloid cells such as monocytes, macrophages and DCs [45]. Removal of extracellular calcium, inhibition of CaSR with the selective antagonist NPS2143 and inhibition of Gα proteins with BIM46187 all inhibit constitutive macropinocytosis in primary hMDMs [45]. Based on these findings, a model for constitutive macropinocytosis has been proposed [45]. Extracellular calcium activates CaSR, allowing Gα to activate PI3K which in turn generates PI(3,4,5)P_3_ at the plasma membrane. PI(3,4,5)P_3_ then recruits phospholipase C (PLC), leading to hydrolysis of PI(4,5)P_2_ to diacylglycerol (DAG) and inositol triphosphate (IP_3_). DAG is phosphorylated by diacylglycerol kinase (DGK) to generate phosphatidic acid, which drives actin polymerisation, constitutive membrane ruffling and constitutive macropinocytosis. Phosphatidic acid drives actin polymerisation by a number of mechanisms including the recruitment of TIAM1, a Rac guanine nucleotide exchange factor (GEF), to the plasma membrane that activates Rac, which is essential for actin polymerisation [46]. While phosphatidic acid is a key driver of constitutive macropinocytosis, phosphatidic acid is also implicated in phagocytosis [57], highlighting the common machinery utilised by both pathways.

##### Membrane Ruffling, Macropinocytosis and Signalling in Macrophages

The peripheral and dorsal ruffling pathways that lead to macropinocytosis have recently been characterised in CSF-1-stimulated BMDMs. Figure 2 shows an example of dorsal ruffles following CSF-1-stimulation of BMDMs by scanning electron microscopy. SNX5, a member of the sorting nexin family of proteins which bind to phosphoinositides via their phox (PX) domains, is a key regulator of macrophage macropinocytosis [17]. SNX5-deficient BMDMs stimulated with CSF-1 show a 60–70% reduction in macropinocytic activity compared to their wild-type counterparts [9]. CSF-1 is able to increase peripheral, but not dorsal ruffling in SNX5-deficient BMDMs, demonstrating that SNX5 is essential for dorsal ruffling but not peripheral ruffling in macrophages [9]. In addition, knockdown of SNX5 in primary BMDMs resulted in smaller macropinosomes (1–2 μm in diameter) than wild-type BMDMs (up to 4 μm in diameter), suggesting that dorsal ruffling generates larger macropinosomes [17]. Together, these data suggest that dorsal ruffling is the major pathway of macropinocytosis in CSF-1-stimulated BMDMs [9]. A model for the role of SNX5 in dorsal ruffling has been proposed [9]: CSF–1 receptor activation leads to increased PI(3,4)P_2_ and SNX5 recruitment to the plasma membrane where it can promote actin polymerisation and dorsal ruffling. Interestingly and in contrast to primary macrophages, SNX5 deficiency does not affect macropinocytosis in immature splenic DCs, implying that different mechanisms of macropinocytosis are utilised by these two immune cells [9].

The mechanism of dorsal ruffle formation in macrophages has been visualised in LPS-activated RAW 264.7 cells using LLSM [54]. The analysis by high-resolution LLSM revealed a novel mechanism to form macropinosomes and also accounted for the large macropinosomes derived from dorsal ruffles. Two filamentous actin (F-actin)-containing filopodial-like extensions (“tent poles”) are erected and subsequently elevate a sheet of F-actin, thereby forming a nascent membrane ruffle. The tent poles circularise to form a circular membrane ruffle. The tent poles cross over each other, constricting the ruffle and causing it to sink into the cell, thus leading to internalisation of a large nascent macropinosome. Rab13 is essential for the generation of the LPS-stimulated tent pole dorsal ruffles and macropinosome formation [54]. Thus, it is clear that there are two distinct macropinocytosis pathways in macrophages which arise from distinct membrane ruffling events and which utilise distinct mechanisms to generate the macropinosomes from these ruffles.

Other machinery has been implicated in macrophage macropinocytosis. Inhibition of PI3K with wortmannin leads to a reduction in macropinocytosis in BMDMs [19]. Peripheral and dorsal ruffles still form in wortmannin-treated BMDMs but do not close to form macropinosomes; rather, they recede back into the cytoplasm, showing that PI3K is required for macropinosome closure [19]. Phosphoinositides are also key regulators of macropinocytosis, as described in a number of reviews [49,58]. The sequential accumulation and breakdown of phosphoinositide species, as well as the role of small GTPases in macropinocytosis, have been mapped in CSF-1-stimulated BMDMs and the maturation process defined [59].

The dorsal, circular, membrane ruffles of activated macrophages are also intimately associated with signalling events. Diffusion within the membranes of the circular ruffles and macropinocytic cups has been shown to be limited, which may provide a mechanism to concentrate lipids and proteins involved in signalling cascades, including those that drive macropinocytosis [59,60]. Macropinocytic cups have been proposed to serve as signalling platforms that accumulate PI(3,4,5)P_3_ in response to growth factor stimulation, which in turn drives activation of Akt and the mTORC1 complex [61]. The dorsal ruffles of activated macrophages have also been shown to be the site for TLR signalling events which regulate cytokine secretion [62,63]. Therefore, the dynamic dorsal ruffles and macropinocytic cups can be considered to be important signalling hubs.

##### Effect of Macrophage Functional State on Macropinocytosis

Following activation, macrophages can differentiate into different functional states defined by their profile of secreted cytokines. For instance, pro-inflammatory macrophages (classically activated, M1) kill pathogens and present antigens to the adaptive immune system, whereas anti-inflammatory macrophages (alternatively activated, M2) subsequently clear inflammation and repair tissue damage [48]. While anti-inflammatory hMDMs show substantial macropinocytosis, pro-inflammatory hMDMs show very little [48]. The low level of macropinocytosis in pro-inflammatory macrophages is attributed to reduced CaSR signalling leading to reduced PI3K activity compared with anti-inflammatory macrophages [48]. As a result, there is decreased production of plasma membrane PI(3,4,5)P_3_ and consequently decreased recruitment of GEFs that activate GTPases required for macropinocytosis (e.g., TIAM1 for Rac1) [48]. Similar to activated DCs, it has been suggested that antigen-presenting pro-inflammatory macrophages may downregulate macropinocytosis to optimise antigen presentation as uptake of antigens is no longer necessary [48].

##### Macropinosome Maturation in Macrophages

The maturation pathway of macropinosomes in CSF-1-stimulated BMDMs has been described [18]. Newly formed macropinosomes are positive for the transferrin receptor (TfR), a marker for early endosomes. The macropinosomes then mature, losing TfR but acquiring Rab7 and lgp-A, markers of late endosomes and lysosomes, respectively. The maturing macropinosomes lose Rab7 but continue to acquire lgp-A, before fusing with tubular lysosomes. During this process, the maturing macropinosomes shrink and migrate centripetally towards the nucleus. Macropinosome shrinkage in CSF-1-stimulated BMDMs can be attributed to the removal of membrane into tubular structures that mediate recycling of macropinocytosed ligands (e.g., albumin) back to the cell surface [21] (Figure 1B). In addition to its role in dorsal ruffling, SNX5 also appears to play a role in the tubulation of macropinosomes [64].

##### Functions of Macropinocytosis in Macrophages

Macropinocytosis is critical to a variety of macrophage physiological processes. As for DCs, macropinocytosis serves as a key pathway of antigen uptake. The more potent amiloride analogue dimethyl amiloride blocks both constitutive macropinocytosis and phorbol myristate acetate (PMA)-stimulated ovalbumin antigen presentation on MHC class I molecules [15]. Constitutive macropinocytosis also contributes to the sentinel function of macrophages by facilitating uptake and delivery of pattern-recognition receptor ligands (e.g., NOD2 ligands) to their intracellular receptors [45]. Macropinocytosis is also the major pathway for internalisation of plasma immunoglobulin G (IgG) and albumin by BMDMs; these internalised plasma proteins are subsequently recycled from newly formed macropinosomes back to the cell surface in a pathway dependent on the neonatal Fc receptor (FcRn) [21]. Macrophage macropinocytosis is also implicated in disease and infection. Macropinocytosis contributes to low-density lipoprotein (LDL) uptake and cholesterol accumulation in hMDMs and is therefore relevant to the development of atherosclerosis [56]. In addition, macrophages are a target of human immunodeficiency virus 1 (HIV-1) infection and HIV-1 is internalised into large vesicles resembling macropinosomes (200 to >500 nm in diameter) in hMDMs [65]. Dimethyl amiloride treatment was able to inhibit HIV-1 replication, suggesting a key role for macropinocytosis in the propagation of the virus [65].

#### 3.1.3. B Cells and T Cells

B cells not only differentiate into antibody-producing plasma cells, but can also act as antigen-presenting cells by internalising antigens through their B cell receptors (BCRs), and by processing the internalised antigens and presenting the processed peptides on MHC II molecules to T cells [66]. While B cells have not previously been considered candidates for active macropinocytosis, a recent study has indicated that B cells are capable of macropinocytosis in the context of pathogen infection. Raji cells, a human B lymphoblast cell line, exhibits very little membrane ruffling and macropinocytosis at rest [67]. However, upon PMA treatment or infection of Raji cells with *Salmonella typhimurium*, *Mycobacterium tuberculosis* or *Mycobacterium smegmatis*, membrane ruffling and macropinocytosis is upregulated [67]. *Mycobacterium tuberculosis* and *Mycobacterium smegmatis* supernatants are also able to induce membrane ruffling and macropinocytosis, suggesting that secreted soluble bacterial factors may be responsible [67].

CD8^+^ T cells directly kill pathogen-infected cells and CD4^+^ T cells secrete cytokines that help the activity of other cells (e.g., B cells and macrophages) to generate an effector immune response. As for B cells, T cells have not traditionally been considered candidates for active macropinocytosis. However, recently, both primary mouse and human CD4^+^ and CD8^+^ T cells have been shown to exhibit significant basal levels of macropinocytosis and, moreover, the basal levels are increased upon cell activation [68]. Treatment with EIP-amiloride not only blocks macropinocytosis by non-activated and activated T cells, but also reduces mTORC1 activation and T cell growth [68]. These findings suggest a model whereby macropinocytosis mediates uptake of amino acids required for activation of mTORC1 and as a result, growth of T cells [68]. Macropinocytosis has also been implicated in the uptake of single-stranded antisense oligonucleotides flanked by locked nucleic acids (LNAs), called GapmeR, to successfully knockdown genes in primary human T cells [69]. In addition, the internalisation of lentiviral vectors pseudotyped with measles virus envelope glycoproteins into unstimulated human peripheral blood T cells appears to be via macropinocytosis based on inhibition of entry with known inhibitors such as amiloride [70]. These recent findings highlight key roles for macropinocytosis in T cell responses and also the infection of T cells.

### 3.2. Endothelial and Epithelial Cells

Various endothelial and epithelial cells have been used to study growth factor-induced macropinocytosis. Both of these cell types are polarised with apical and basolateral plasma membrane domains separated by tight junctions. Endothelial cells line blood and lymphatic vessels, with the apical surface facing the lumen of the vessels, whereas epithelial cells line surfaces of the body, such as lung, small intestine and urinary tract, with the apical surface facing the external environment. Although the majority of studies exploring macropinocytosis in these cells have utilised immortalised cell lines, they have nonetheless provided valuable insights into molecular mechanisms of macropinocytosis.

#### 3.2.1. Endothelial Cells

Macropinocytosis is relevant to various aspects of endothelial cell biology. These cells exhibit low levels of constitutive macropinocytosis, but can upregulate macropinocytosis in response to growth factors. For example, cultured porcine aortic endothelial (PAE) cells show low levels of basal membrane ruffling but upregulate PI(3,4,5)P_3_ levels and membrane ruffling in response to platelet-derived growth factor (PDGF) [20]. PDGF-stimulated PI(3,4,5)P_3_ production and membrane ruffling in these cells is dependent on PI3K [20].

Endothelial cells can also be the natural infection targets of viruses, which can exploit a variety of endocytic pathways for uptake, including macropinocytosis (reviewed in [6]). For example, HIV-1 enters primary human brain microvascular endothelial cells (BMVECs) via a pathway of macropinocytosis that appears to require cholesterol and MAPK signalling [71]. The majority of internalised HIV-1 particles are then trafficked to lysosomes for degradation [71]. Entry of Kaposi’s sarcoma-associated herpesvirus (KSHV) into primary human microvascular dermal endothelial cells (HMVEC-d) and human umbilical vein endothelial cells (HUVECs) is also via macropinocytosis [72]. KSHV infection significantly increases membrane ruffling and 70 kDa dextran uptake in both cell types, whose levels are low in the absence of infection [72]. Invasion of the bacterium *Escherichia coli* K1 into human brain microvascular endothelial cells (HBMECs) is also via macropinocytosis [73].

Macropinocytosis is also relevant to key cellular processes in endothelial cells including regulation of gene expression and metabolism. The gene expression profile of hCMEC/D3 cells, a human brain microvascular endothelial cell line that is a model for the blood–brain barrier, is altered by the uptake of neutrophil-derived microvesicles [74]. Macropinocytosis contributes to the cellular uptake of these microvesicles based on a reduction in uptake on treatment with either EIP-amiloride, cytochalasin D or wortmannin, although the clathrin-mediated endocytosis inhibitor monodansylcadaverine also reduces uptake [74]. HUVECs also upregulate macropinocytosis on glutamine deprivation to facilitate the uptake of non-essential amino acids and to drive proliferation [75]. One of the limitations in the study of endothelial cells is the lack of primary cell cultures. The recent advance in generation of human primary endothelial cells from blood (blood outgrowth endothelial cells) [76] should provide a valuable source of primary endothelial cells in the future for detailed analysis of the macropinocytosis pathway in this polarised cell.

#### 3.2.2. Epithelial Cells

Madin–Darby Canine Kidney (MDCK) cells have proven to be a useful cell line to investigate the effect of cell polarity on macropinocytosis. From these studies, it is clear that the apical and basolateral surfaces of polarised epithelial cells differ in their levels of macropinocytosis. The apical surface of polarised MDCK cells shows membrane ruffling and macropinocytosis in response to N-ethylmaleimide treatment but not the basolateral surface [77]. The resulting macropinosomes from the apical surface remain in the apical cytoplasm and do not fuse with other endocytic organelles [77]. The viral oncoprotein v-Src also increases the uptake of HRP from the apical but not basolateral surface in MDCK cells, which is attributed to the induction of apical ruffling and macropinocytosis [78]. The resulting apical macropinosomes, in contrast to N-ethylmaleimide-induced apical macropinosomes, are thought to fuse with the apical recycling endosome [78].

In addition to MDCK cells, a wide variety of other epithelial cells have been used to interrogate the mechanisms of macropinocytosis and its physiological relevance. A431 human epidermoid carcinoma cells are one of the most commonly used epithelial cell lines to study macropinocytosis. Basal levels of macropinocytosis in these cells is low, but can be increased upon the addition of epidermal growth factor (EGF) [11]. Macropinosomes induced by EGF stimulation avoid trafficking to transferrin-labelled early and recycling endosomes and LDL-labelled late endosomes and lysosomes [79] and instead recycle to the plasma membrane [15]. These EGF-stimulated macropinosomes are capable of tubulating and fusing with one another [79]. The use of EGF-stimulated A431 cells has also identified the CtBP1/BARS protein as a regulator of macropinosome closure [80] and elucidated the mechanism by which amiloride inhibits macropinocytosis [30]. Various stimulators of macropinocytosis have been identified in other epithelial cells. For example, binding of arginine-rich peptides to cell surface proteoglycans leads to Rac1 activation and macropinocytosis in the Chinese hamster ovary (CHO) cell line [81] and lipids bound to albumin such as free fatty acids stimulate macropinocytosis in immortalised podocytes, epithelial cells which form the filtration barrier of the kidney [82].

Like endothelial cells, epithelial cells are also natural infection targets of numerous pathogens including bacteria and viruses, which can exploit macropinocytosis to gain entry (Figure 1C). A549 is a human lung epithelial cell line with low levels of ruffling and macropinocytosis; infection with *Mycobacterium smegmatis* or *Mycobacterium tuberculosis* increases membrane ruffling and macropinocytosis [83]. Incubation with the *Mycobacterium* supernatants also increases macropinocytosis, suggesting that secreted soluble bacterial factors are responsible for the induction of macropinocytosis [83]. Macropinocytosis is also exploited by other bacteria for internalisation, such as the entry of invasive *Salmonella typhimurium* into HEp-2 epithelial cells [84] and the entry of *Neisseria gonorrhoeae* into primary human urethral epithelial cells (HUECs) [85]. Various viruses also exploit macropinocytosis to mediate their entry into epithelial cells. Ebola virus uses dynamin-dependent macropinocytosis to enter Vero cells, a monkey kidney epithelial cell line [86] and echovirus 1 uptake into polarised Caco-2 cells, a human colon cancer cell line, depends on macropinocytosis [87]. In addition, different pathways of macropinocytosis can be utilised by viruses to gain entry into epithelial cells. Western Reserve (WR) and International Health Department-J (IHD-J) are closely related strains of the Vaccinia virus, and both stimulate macropinocytosis to facilitate their internalisation into HeLa cells, a human cervical cancer epithelial cell line [88]. However, infection with WR Vaccinia virus is sensitive to PI3K inhibition whereas infection with IHD-J Vaccinia virus is not [88]. Furthermore, infection with WR Vaccinia virus is highly dependent on Rac1, whereas infection with IHD-J is highly dependent on Cdc42, highlighting the existence of different pathways of macropinocytosis that can be exploited by viruses [88]. Spherical influenza virus can enter cells by both macropinocytosis and by clathrin-mediated endocytosis whereas the larger filamentous influenza virus is likely to enter epithelial cells mainly by macropinocytosis [89,90]. Thus, macropinocytosis is an entry portal for many infectious pathogens that invade epithelial layers.

### 3.3. Fibroblasts

Fibroblasts are found in connective tissue and synthesise and secrete key extracellular matrix molecules such as collagen. Fibroblast cell lines have been frequently used as a cell model to study macropinocytosis and to identify the relevant molecular regulators. For example, the importance of Rac1 to macropinocytosis was demonstrated in these cells. Microinjection of activated Rac1 into Rat2 fibroblasts, which have low levels of constitutive macropinocytosis, drastically increased bulk-phase pinocytosis which was considered to be due to macropinocytosis [91]. Similarly, microinjection of activated Rac1 into 3T3 mouse fibroblasts, which exhibit low levels of membrane ruffling, dramatically increases membrane ruffling [91]. Peripheral and dorsal ruffling pathways have also been studied in fibroblasts. Unstimulated AG 1523 human foreskin fibroblasts show little to no ruffling, but can form circular dorsal ruffles in response to PDGF [92]. It has been shown that PAK1, a kinase activated by Rac1 and Cdc42, plays a key role in PDGF-stimulated macropinocytosis in 3T3 fibroblasts [93]. The role of WAVE proteins, which are regulators of actin polymerisation, in peripheral and dorsal ruffling has been identified through studies of PDGF-stimulated mouse embryonic fibroblasts [94]. WAVE proteins regulate ruffling through activation of the Arp2/3 complex, which nucleates actin polymerisation [94]. WAVE1 is required for dorsal ruffling whereas WAVE2 is required for peripheral ruffling [94].

### 3.4. Cells of the Nervous System

There have only been a handful of studies exploring macropinocytosis in cells of the nervous system. However, these studies using primary neurons, neuron-like cell lines, primary microglia and microglia cell lines, have all suggested a significant role for macropinocytosis in the physiology of these cells as well as in the context of disease, and macropinocytosis represents an important emerging topic in the neuroscience field.

#### 3.4.1. Neurons

Macropinocytosis has been proposed to play a role in the spread of pathogenic protein aggregates from neuron to neuron and the propagation of neurodegenerative diseases such as Alzheimer’s disease, Parkinson’s disease and Huntington’s disease [95]. In C17.2 cells, a mouse neural precursor cell line, tau fibrils, which represent one of the hallmarks of Alzheimer’s disease, can be internalised by a macropinocytosis-like pathway, based on several lines of evidence including internalisation into large vesicles (>5 μm in diameter) and a decrease in uptake following treatment with either EIP-amiloride or cytochalasin D [96]. Interestingly, this pathway of tau fibril uptake is dependent on binding to cell surface heparan sulfate proteoglycans (HSPGs); the existence of a HSPG-dependent macropinocytic pathway is supported by studies of primary mouse hippocampal neurons, in which depletion of *Ext1*, an enzyme involved in HSPG synthesis, reduces uptake of tau fibrils but not transferrin [96]. In contrast, another study showed that the uptake of tau aggregates by human induced pluripotent stem cell (iPSC)-derived cerebral cortex neurons was not inhibited by cytochalasin D [97], indicating that different pathways for internalisation of protein aggregates may be used in different neuronal model systems. One of the pathogenic aggregates identified in amyotrophic lateral sclerosis (ALS), namely superoxide dismutase 1 (SOD1), is internalised by macropinocytosis by both NSC-34 cells, a mouse motor neuron-like cell line, and human iPSC-derived motor neurons [98]. In both cell models, SOD1 aggregates stimulate ruffling and the uptake of 10 kDa dextran [98]. Macropinocytosis is also potentially implicated in the aetiology of Alzheimer’s disease through the regulation of amyloid precursor protein (APP) processing. SN56 cells, a hybrid cell line made from fusing dissociated embryonic mouse septal neurons with N18TG2 neuroblastoma cells, internalise APP from the cell surface into large vesicles (~1 μm in diameter) which traffic to lysosomes, raising the possibility that macropinocytosis may be involved in the regulation of a pathway that leads to APP cleavage and generation of toxic amyloid β peptides [99]. Dominant negative Arf6 reduces the internalisation of cell surface APP and decreases amyloid β production, suggesting that Arf6 may regulate macropinocytosis of APP [99].

Macropinocytosis also appears to regulate the dynamics of the growing or regenerating ends of neuronal axons, called growth cones. Caffeine treatment has been used to study axon growth inhibition and growth cone collapse, defined as a reduction in the surface area of a growth cone [100]. Caffeine-stimulation is associated with axon growth inhibition and growth cone collapse and the formation of large vesicles which can be loaded with the fluid-phase marker, 10 kDa dextran, that are trafficked back to the cell body [100]. These studies suggest that macropinocytosis promotes membrane internalisation at growth cones and mediates growth cone collapse and axon growth inhibition [100]. Semaphorin 3A (collapsin-1) has been identified as a key mediator of macropinocytosis-induced axon growth inhibition [100,101,102] by facilitating the degradation of syntaxin 1B, a negative regulator of macropinocytosis [101]. Further support for a the role of macropinocytosis in growth cone collapse comes from the finding that the dominant negative form of Rac1 inhibits semaphorin 3A-mediated growth cone collapse and axon growth inhibition [102]. Macropinocytosis could also be relevant in the context of neuron injury. Spinal cord injury causes neuron growth cones to become swollen and distorted [103]. In vitro and in vivo analyses have shown that these axon endings are highly dynamic, extending sheets of membrane resembling membrane ruffles, containing large (2–3 μm) intracellular vesicles and internalising 10 kDa dextran, suggesting that these are sites of active macropinocytosis. However, the precise role of macropinocytosis in the pathology of spinal injury and/or repair processes remains unclear. Nonetheless, these studies suggest that macropinocytosis is important for the maintenance of axons, and therefore potentially the formation of synaptic connections.

#### 3.4.2. Microglia

Microglia are considered the “macrophages” of the brain and play key roles in immunity [104]; as such, they are good candidates for active macropinocytosis. This is supported by the observation that resting rat microglia exhibit high levels of Lucifer yellow uptake both in vitro and in vivo, indicating that these cells have high levels of pinocytosis, to which macropinocytosis may contribute [105]. The uptake of amyloid β by the macropinocytosis pathway in microglia is a key consideration for the development of Alzheimer’s disease. BV-2 cells, an immortalised mouse microglia cell line, and primary mouse microglia in vitro and in vivo can internalise soluble amyloid β [106]. For BV-2 cells, the uptake of soluble amyloid β is considered to be mediated via a macropinocytosis-like mechanism, as the uptake is nonsaturable and sensitive to cytochalasin D treatment [106]. Internalised soluble amyloid β is trafficked to lysosomes for degradation [106]. Fluid-phase uptake of soluble amyloid β by microglia is enhanced by various stimuli. For example, addition of nerve growth factor (NGF) to mouse primary microglia increases macropinocytic uptake of 70 kDa dextran but not phagocytosis of 6 μm beads, and increases uptake of soluble amyloid β but not fibrillar amyloid β [107]. Macropinocytosis is also relevant to the uptake of exosomes by microglia. The mouse microglial cell line EOC-20 can internalise exosomes in a process involving membrane ruffling and which is sensitive to amiloride and cytochalasin D, suggesting that macropinocytosis is also a key pathway of exosome uptake in microglia [108]. It is clear that macropinocytosis by microglia, as for macrophages, represents an active pathway. However, an understanding of the physiological relevance of macropinocytosis by microglia in immune protection and in disease development in the central nervous system requires further investigation.

### 3.5. Cancer Cells

Cancer cells are characterised by abnormal proliferation and by invasion into tissues in the body. As a result of malignant invasion, tumours are often poorly vascularised and hypoxic, and therefore require adaptive mechanisms to facilitate nutrient acquisition for sustained proliferation [109]. These adaptive mechanisms include rewiring of metabolic pathways (e.g., increased utilisation of glycolysis, i.e., the Warburg effect) and also the utilisation of macropinocytosis to facilitate nutrient uptake in nutrient-poor environments. While certain aspects of macropinocytosis in cancer cells have already been described in earlier sections, the role of macropinocytosis in nutrient acquisition by cancer cells has been more extensively interrogated in other cancer cells. The role of macropinocytosis in cancer cells and the therapeutic potential of targeting macropinocytosis in the treatment of cancer have been recently reviewed in detail [110,111,112]. Here, we give a few examples as follows.

Compared to cells of normal tissue, human pancreatic ductal adenocarcinomas (PDACs) show increased levels of macropinocytosis, as indicated by increased internalisation of high-molecular-weight dextran [109]. Furthermore, the mouse pancreatic cancer cell line KRPC can derive amino acids from the proteolytic degradation of extracellular albumin to sustain their proliferation [109]. The authors showed that these adenocarcinoma cells were able to proliferate in the absence of essential amino acids in the medium as a consequence of lysosomal degradation of extracellular albumin. The acquisition of these amino acids from extracellular albumin is inhibited by EIP-amiloride, suggesting that macropinocytosis is used by pancreatic cancer cells to scavenge extracellular proteins such as albumin to fuel proliferation [109]. The in vivo relevance of macropinocytosis by cancer cells is also demonstrated by analysis of the growth of tumour xenografts. Mouse xenografts of the human pancreatic adenocarcinoma cell line (MIA PaCa-2) showed reduced high-molecular-weight dextran uptake after treatment of mice with EIP-amiloride and, in addition, tumours were smaller compared to controls [16], suggesting that inhibition of macropinocytosis in vivo may be a viable option for the treatment of susceptible cancers. As macropinocytosis is nonspecific, this pathway can also facilitate the uptake of other molecules which mediate cancer cell proliferation and survival, for instance, uptake of ATP by A549 human lung epithelial cancer cells, as demonstrated by the co-localisation of fluorescent ATP and the 70 kDa dextran marker of macropinocytosis [111]. Notably, inhibition of macropinocytosis with EIP-amiloride increases the sensitivity of A549 cancer cells to the ATP competitive inhibitor sunitinib [113].

The induction of constitutive macropinocytosis by oncogenic Ras appears to be a key feature of numerous cancer cell types [16,109,113] (Figure 1D). In addition to sustaining the proliferation of cancer cells through the acquisition of amino acids, macropinocytosis of extracellular nutrients can also act as a sensor to drive the activation of the mTORC1 complex in mouse embryonic fibroblasts (MEFs) expressing oncogenic K-Ras [114]. Following starvation of essential amino acids, extracellular albumin promoted higher mTORC1 activity in MEF cells expressing oncogenic K-Ras compared with wild-type K-Ras, a process inhibited by EIP-amiloride [112]. mTORC1 is a major signalling complex that promotes cell proliferation [115]. Hence, constitutive macropinocytosis induced by oncogenic Ras may be a mechanism of sustained mTORC1 activation and cell proliferation in cancer cells [115]. Aside from oncogenic Ras, various other pathways can drive macropinocytosis in cancer cells. A recent shRNA screen using three bladder cancer cell lines expressing wild-type Ras identified the canonical Wnt signalling pathway, and transcription of β-catenin target genes, as a driver of macropinocytosis [116]. In addition, oncogenic Src family kinases, involved in numerous cancers, induce macropinocytosis [117,118]. Macropinocytosis is not only involved in cancer cell growth, but is also implicated in cell death of aggressive gliobastoma cancer cells. Expression of constitutively active H-Ras in U251 glioblastoma cells triggers the formation of abundant macropinosomes, resulting in reduced viability and cell death [119]. This form of cell death associated with increased macropinocytosis is nonapoptotic and termed “methuosis” [119]. Constitutively active H-Ras also triggers enhanced macropinocytosis and methuosis in several other human glioma cell lines, but not transformed HeLa, HEp2 and HEK293 cells, demonstrating different requirements for the induction of macropinocytosis in different cell types [119].

## 4. Conclusions

Macropinocytosis is critical to an ever-growing number of cellular processes, including but not limited to antigen presentation, signalling, nutrient acquisition, nutrient sensing and pathogen invasion. While macropinocytosis has been well studied in dendritic cells and macrophages, it is also becoming increasingly apparent that macropinocytosis occurs in, and is physiologically relevant to, a much wider range of cell types, including cells of the adaptive immune system, polarised endothelial and epithelial cells, cells of the nervous system and cancer cells. Studies of these systems have revealed not only the diversity of macropinocytosis-dependent functions, but also the existence of different macropinocytic pathways that are initiated under different conditions and stimuli and that are regulated by different molecular machinery. The identification of two distinct macropinocytosis pathways in macrophages, derived from distinct types of membrane ruffling, namely dorsal and peripheral ruffling, highlights the complexity of these pathways, which in turn reflect distinct functional attributes. Clearly, a deeper understanding of the stimuli, regulators and the fate of the macropinocytic pathways in different cell types is now required to fully appreciate the underlying cell biology and physiology.

In spite of recent advances, our understanding of the function and mechanism of macropinocytosis in different cell types is still in its infancy. This is partly due to insufficient information on macropinocytosis in primary cells, but also due to the absence of macropinosome-specific markers and the absence of drugs that can specifically modulate macropinocytosis both in vitro and in vivo. Identification of novel molecular machinery regulating macropinocytosis will aid future studies and may pave the way to specifically modulate macropinocytosis in vitro and in vivo. With the application of small-molecule inhibitors [55], siRNA libraries [120] and CRISPR libraries, there are exciting opportunities for high-throughput screens to be performed to identify novel macropinocytosis regulators. Ultimately, a greater understanding of the different pathways of macropinocytosis in different cell types will provide the potential to modulate individual functions which are regulated by these endocytic pathways.

## Figures and Tables

**Figure 1 membranes-10-00177-f001:**
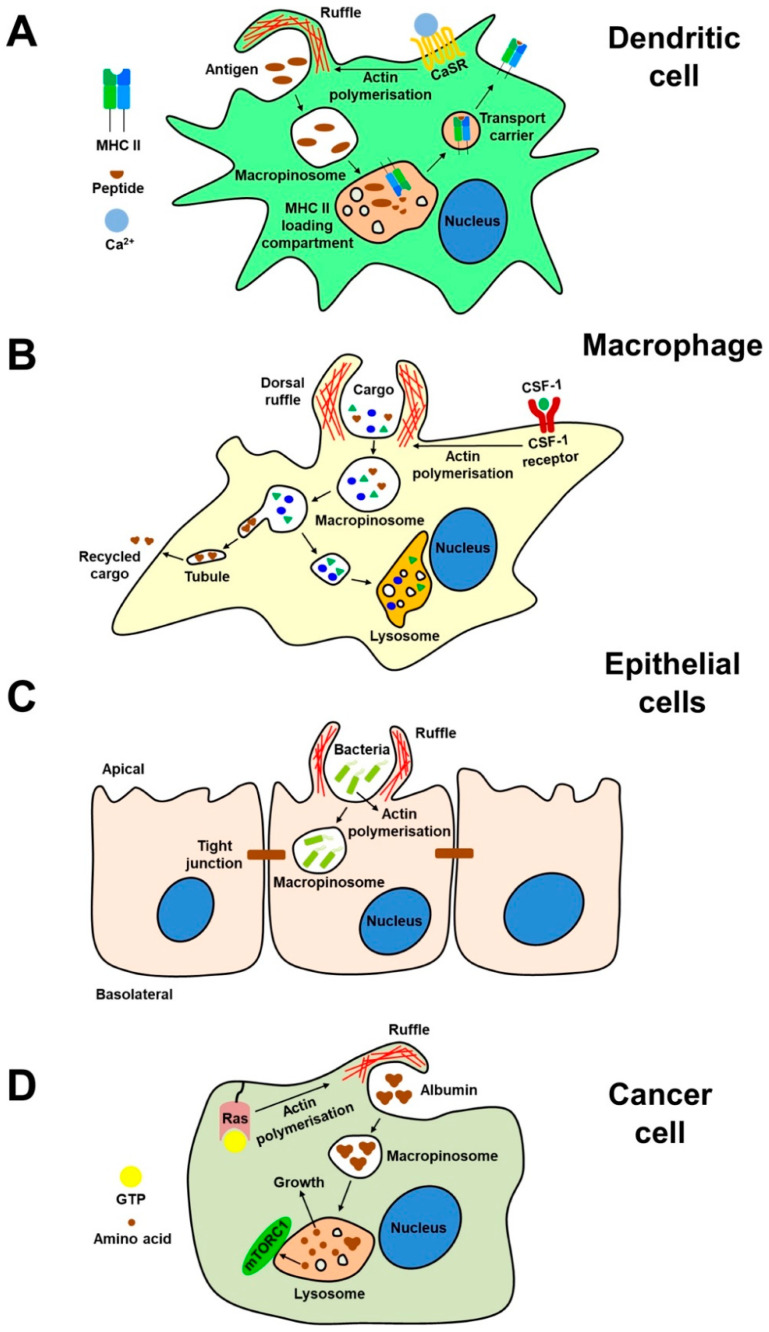
Macropinocytosis in different cell types. (**A**). Dendritic cells use constitutive macropinocytosis to internalise antigens that are subsequently trafficked to MHC II-positive compartments. (**B**). Macrophages can internalise substantial amounts of extracellular fluid and solutes in response to growth factor stimulation, which are trafficked to lysosomes or alternatively recycled to the cell surface via tubular carriers. (**C**). Epithelial cells show low levels of macropinocytosis at rest, but pathogens such as bacteria can induce macropinocytosis to facilitate their internalisation. (**D**). Cancer cells expressing oncogenic Ras constitutively macropinocytose, which facilitates the uptake of extracellular nutrients which activate mTOR signalling and drive growth and proliferation.

**Figure 2 membranes-10-00177-f002:**
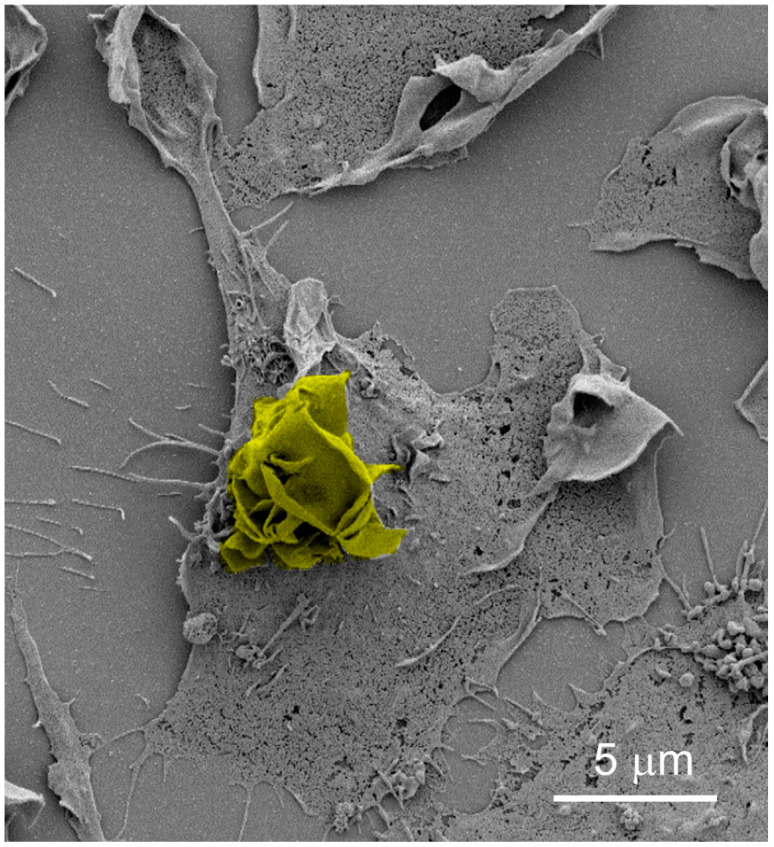
Dorsal ruffles on activated macrophages. Scanning electron micrograph of dorsal plasma membrane ruffles (pseudocoloured yellow) in CSF-1-activated bone marrow-derived macrophages. Deficiency of SNX5 results in a dramatic reduction in dorsal ruffling and macropinocytosis. Micrograph taken by Dr Prajakta Gosavi from article by Lim et al. [9] (3500× magnification).

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
