# Peer review of "Macropinocytosis in Different Cell Types: Similarities and Differences"

_membranes, 2020, doi:10.3390/membranes10080177_

Round 1

Reviewer 1 Report

This is a well-written and comprehensive review regarding macropinocytosis in different cell types.  The authors described the diverse roles of macropinocytosis process in cells of multiple systems, such as immune systems, nerve systems, polarized endothelial and epithelial cells, even in cancer cells.

This manuscript generally would be suitable to be published, but I recommend the authors would address more physiological relevance in distinct cellular systems, especially cancer cells systems which needs to introduce more experimental details and results for explaining the physiological significance of this process.

Author Response

Please see uploaded PDF file

Reviewer 2 Report

The authors review macropinocytosis in different cell types and it is clear that they do intend to provide detailed general mechanisms however I think that still some should be provided to introduce the subject such as the potential role of annexins and Sorting of membrane components from endosomes and subsequent recycling to the cell surface occurs by a bulk flow process.

S Mayor, JF Presley, FR Maxfield Journal of Cell Biology 121 (6), 1257-1269   I also think that this review would benefit from some discussion of new insights into the role of galectin 3   

Author Response

I did not see this review until i was about the resubmit and wondered if it had been added later.

Regardess, I thank the reviewer for his/her comments. The point raised by the reviewer on annexins and sorting nexins is not particularly relevant to the central element of the research topic.  The relationship of endosomal sorting, as mentioned by the reviewer, and recycling from macropinosomes remains unclear in the literature, hence the components used by early endosomes may or may not be relevant. We have discussed the role of sorting nexin 5 in the review. Galectin 3 is not relevant to our topic under discussion

The paper mentioned by the reviewer is very old (1993) and I do not directly relevant to the topic under discusion

Reviewer 3 Report

In this review, Lin and colleagues nicely summarised the current and recent knowledge gained on the molecular mechanisms underlying macropinocytosis in a broad range of cell lines.

The review is well organised and pleasant to read.

The topic is complex and highly cell type-dependent; therefore, this review offers a good summary and overview of the similarities and differences of macropinocytosis in different cellular contexts.

Author Response

We thank the reviewer for her/his comments. We note that no changes or amendments were requested.